# Efficacy of Organ Preservation Strategy by Adjuvant Chemoradiotherapy after Non-Curative Endoscopic Resection for Superficial SCC: A Multicenter Western Study

**DOI:** 10.3390/cancers15030590

**Published:** 2023-01-18

**Authors:** Arthur Berger, Guillaume Perrod, Mathieu Pioche, Maximilien Barret, Elodie Cesbron-Métivier, Vincent Lépilliez, Marianne Hupé, Enrique Perez-Cuadrado-Robles, Franck Cholet, Augustin Daubigny, Charles Texier, Einas Abou Ali, Edouard Chabrun, Jérémie Jacques, Timothee Wallenhorst, Jean Baptiste Chevaux, Marion Schaefer, Christophe Cellier, Gabriel Rahmi

**Affiliations:** 1CHU Bordeaux, Department of Gastroenterology and Digestive Endoscopy, Univ. Bordeaux, 33000 Bordeaux, France; 2Department of Gastroenterology and Digestive Endoscopy, Georges Pompidou European Hospital, Paris-Cité University, 75015 Paris, France; 3Department of Hepatology and Gastroenterology, Edouard Herriot Hospital, Lyon 1 University Claude Bernard, 69003 Lyon, France; 4Department of Gastroenterology and Digestive Endoscopy, Cochin University Hospital, University of Paris, 75014 Paris, France; 5Department of Hepatology and Gastroenterology, Angers Teaching Hospital, Angers University, 49000 Angers, France; 6Department of Hepatogastroenterology, Jean Mermoz Private Hospital, Ramsay Santé, 69008 Lyon, France; 7Department of Hepatology and Gastroenterology, Brest Teaching Hospital, Brest University, 29200 Brest, France; 8Department of Hepatology and Gastroenterology, Limoges Teaching Hospital, Limoges University, 87000 Limoges, France; 9Department of Hepatology and Gastroenterology, Rennes Teaching Hospital, Rennes University, 35033 Rennes, France; 10Department of Endoscopy and Hepatogastroenterology, Nancy Teaching Hospital, Nancy University, 54000 Nancy, France

**Keywords:** esophageal squamous cell carcinoma, endoscopic resection, adjuvant chemoradiotherapy, organ preservation

## Abstract

**Simple Summary:**

Endoscopic resection (ER) of superficial esophageal SCC is a safe and efficient treatment. Compared with surgery, ER is minimally invasive and associated with lower morbidity and mortality. This is the largest Western study to investigate long-term outcomes after ER for superficial esophageal SCC and to measure the effect of CRT after non-curative resection. The combination therapy of ER and adjuvant CRT reduced the risk of nodal recurrence. This organ preservation strategy may be considered as an alternative to adjuvant surgery for patients with high comorbidities and high risk of nodal invasion after ER of superficial SCC.

**Abstract:**

Background. In case of high risk of lymph node invasion after endoscopic resection (ER) of superficial esophageal squamous cell carcinoma (SCC), adjuvant chemoradiotherapy (CRT) can be an alternative to surgery. We assessed long-term clinical outcomes of adjuvant therapy by CRT after non-curative ER for superficial SCC. Methods. We performed a retrospective multicenter study. From April 1999 to April 2018, all consecutive patients who underwent ER for SCC with tumor infiltration beyond the muscularis mucosae were included. Results. A total of 137 ER were analyzed. The overall nodal or metastatic recurrence-free survival rate at 5 years was 88% and specific recurrence-free survival rates at 5 years with and without adjuvant therapy were, respectively, 97.9% and 79.1% (*p* = 0.011). Independent factors for nodal and/or distal metastatic recurrence were age (HR = 1.075, *p* = 0.031), Sm infiltration depth > 200 µm (HR = 4.129, *p* = 0.040), and the absence of adjuvant CRT or surgery (HR = 11.322, *p* = 0.029). Conclusion. In this study, adjuvant therapy is associated with a higher recurrence-free survival rate at 5 years after non-curative ER. This result suggests this approach may be considered as an alternative to surgery in selected patients.

## 1. Introduction

With more than 500,000 new cases diagnosed in 2018, esophageal cancer represents the eighth most common cancer worldwide [1]. Esophageal cancer is characterized by two histological types: adenocarcinoma and squamous cell carcinoma (SCC). In Europe, SCC are much less frequent than adenocarcinomas, as opposed to Asia, where SCC is the dominant histological type [1]. Independently of histological type, endoscopic resection (ER) by either endoscopic mucosal resection (EMR) or submucosal dissection (ESD) has become the first line treatment for superficial esophageal cancer in the past decades [2]. Recently, 14.2% disease recurrence and 81.8% recurrence-free survival rates at 5 years have been reported after ER for superficial SCC [3]. According to ESGE guidelines, ESD is recommended for tumors > 10 mm because it is associated with a lower recurrence rate than EMR [4]. ER is considered as curative only for tumors with low risk of lymph node invasion. Risk varies mainly depending on tumor invasion depth. For SCC, the risk is significant (8.0–18.0%) when infiltration depth reaches or exceeds the muscularis mucosae (m3). This risk is increased to 11.0–53.1% in the case of submucosal invasion ≤ 200 μm (sm1), and 30.0–53.9% for SCC with deep submucosal invasion (>200 μm) (sm2/3) [5,6,7]. Degree of differentiation, as well as lymphovascular invasion, are also associated with a significant risk of nodal invasion [8,9,10]. Esophagectomy is the recommended therapy for esophageal SCC with risk of nodal invasion [4]. However, surgery is at risk of complications and a substantial part of patients with severe comorbidities may not be eligible [11]. Development and evaluation of new strategies is, therefore, essential for these patients. To date, adjuvant chemoradiotherapy (CRT) remains the less invasive alternative with promising results in the field [12]. Indeed, organ preservation strategy, defined as an association of ER and adjuvant CRT, has shown promising results [13,14,15,16]. Recent studies reported similar or reduced risks of lymph node metastasis for SCC after EMR [17] or ESD [18,19] for patients treated by adjuvant CRT, as compared to surgery. However, these studies lack long-term follow-up, which is a major element of curative strategies evaluation. There is limited data on this topic in Western countries.

To that end, we conducted a retrospective multicenter study to assess the long term oncological outcomes of adjuvant treatment by CRT after non-curative ER of superficial SCC.

## 2. Materials and Methods

### 2.1. Study Population

From April 1999 to April 2018, we performed a retrospective multicenter study, including all consecutive patients treated by ER in 11 French tertiary care hospitals for histologically proven SCC with infiltration beyond the muscularis mucosae. The inclusion criteria were as follows.

Esophageal SCC defined as superficial after endoscopic assessment by white-light and Lugol chromoendoscopy. Absence of lymph node invasion was confirmed by computed tomography (CT) and/or endoscopic ultrasonography (EUS) before ER or by negative microscopic margin (R0) if pathology was not evaluable, for example, due to piecemeal resection and/or coagulation artifacts (Rx). Cases with microscopic margins positive for tumor were excluded. Cases with a follow-up shorter than 6 months were excluded, except for an early nodal or metastatic recurrence before that time. All data regarding clinical, endoscopic, and histological adverse events after adjuvant treatment (esophagitis, fistula, and stenosis) were collected. The study protocol conforms to the ethical guidelines of the 1975 Declaration of Helsinki, as reflected in a priori approval by the institution’s Human Research Committee. The research proposal was approved by the Local Ethics Committee of the Georges-Pompidou European Hospital (CERAPHP.5, IRB registration number: 00011928).

### 2.2. Endoscopic Resection

ER was performed, as previously described by our group [3]. After ER, tumor specimens were spread then fixed in formalin and evaluated by local expert pathologists.

### 2.3. Histological Definitions

Resection in one-piece was defined as en-bloc resection. Complete resection was defined as “R0” if deep and lateral margins were negative for tumor. Tumor infiltration limited to the muscularis mucosae (m3) was classified as pT1a and pT1b for cancer with submucosal invasion of ≤200 μm (sm1) or submucosal invasion > 200 μm (sm2/3) [4]. The TNM status was assessed according to the guidelines of the European Society of Gastrointestinal Endoscopy (ESGE) [4]. According to the ESGE guidelines, expanded curative resection was defined as a well differentiated lesion with infiltration depth of ≤Sm1, and no lymphovascular invasion. Expanded curative resection is associated with a low risk of nodal and/or distal metastasis. A high risk of nodal or distal metastasis was defined as an infiltration depth ≥Sm2, positive lymphovascular invasion, or moderate and poor differentiation. Local recurrence and synchronous cancer were defined, respectively, as positive biopsy of SCC in the ER area, and a new SCC detected in a distinct area from the initial resection.

### 2.4. Objectives and Outcomes

The main objective was to compare long-term clinical outcomes of patients treated with and without adjuvant treatments after non-curative ER for superficial esophageal SCC with tumor infiltration beyond the muscularis mucosae. The secondary objectives were to identify risk factors for nodal or metastatic cancer recurrence and evaluate the impact of adjuvant treatments. The main outcome was the overall nodal or metastatic cancer recurrence-free survival rate at 5 years.

### 2.5. Follow-Up after Endoscopic Resection and Adjuvant Therapy

Follow-up data were obtained from the medical records. Upper-gastrointestinal endoscopy with esophageal biopsies was repeated 2–6 months after ER and annually thereafter. For tumors with a risk of local recurrence (i.e., positive lateral margin) and in case of local recurrence, endoscopic re-treatment was proposed. Nodal and metastatic cancer recurrence were screened by CT and/or EUS performed according to local protocols [20]. A minimal follow-up of 6 months was required for inclusion in the study, or cases of nodal or distal metastasis cancer recurrence or death secondary to cancer before 6 months during the follow-up were included. The end of follow-up was defined either as the date of cancer recurrence or as the date of death or last CT and/or EUS, whichever occurred first. All adjuvant therapies were discussed at an oncological multidisciplinary local meeting. Adjuvant therapy, i.e., surgery or CRT, was proposed depending on patient’s comorbidities. The CRT protocol was in accordance with French guidelines [21]. Radiation treatment was delivered at 1.8–2.0 Gy per fraction, with a total dose of 40–50 Gy over 25–30 fractions. Chemotherapy was chosen between the two following regimens and delivered during radiotherapy at 3-weeks intervals: either 5-fluorouracil (800 mg/m²) administered on days 1 to 5 in association with Cisplatin (100 mg/m²) administered with hydration on day 1, or a 5-fluorouracil (400 mg/m^2^) bolus on day 1 followed by 5-fluorouracil (1600 mg/m^2^) administered over 48 h, in association with Oxaliplatin (85 mg/m^2^) with hydration and folic acid on day 1.

### 2.6. Statistical Analysis

The characteristics of patients and tumors are expressed as medians (range) or frequencies (%). The follow-up period started at the time of EMR or ESD and ended at the date of last follow-up or death. Time to nodal or metastatic recurrence was calculated between the date of ER and that of diagnosis of recurrence. Population characteristics were compared by chi-squared or Fisher’s exact test and t-test, respectively, for qualitative and quantitative variables. Disease-free and overall survival rates were calculated using the Kaplan–Meier method. In univariate analyses, comparisons of tumors with and without nodal or metastatic cancer recurrence were performed using log-rank test and Cox proportional hazards regression model, respectively, for qualitative and quantitative variables. All significant factors, together with those of borderline significance (*p* < 0.2), were included in the multivariate analyses, which used the Cox proportional hazards model. In the multivariate models, only variables with a *p*-value < 0.05 were included and considered indicative of significance. Statistical analysis was conducted using SPSS software version 24.0 (IBM Corp., Armonk, NY, USA).

## 3. Results

### 3.1. Study Population and Tumor Characteristics

One-hundred and thirty-two patients were consecutively enrolled, and one-hundred and thirty-seven tumors were resected (Figure 1). The median age was 63 (range 35–90) years, 39.8% of patients were active smokers, and 51% used alcohol daily. A history of head-and-neck squamous cell carcinoma was noted in 55 patients (41.6%). Five patients (3.8%) had a history of multiple ERs for superficial SCCs. The characteristics of the tumors are summarized in Table 1. The number of procedures per center is listed in Appendix A. A total of 71 superficial esophageal SCCs (51.8%) were treated by EMR (EMR cap in 87.7% and ligation mucosectomy in 12.3%) and 66 (48.2%) by ESD (Appendix A). Complications after ER are detailed in the Appendix A. Period bias assessment and rates of EMR/ESD performed by the tertile period are shown in Appendix A.

### 3.2. Main Outcome

The median follow-up period was 22 months (range 1–135 months) for all cases. Nodal or metastatic recurrence during the first 6 months of follow-up occurred in nine patients. Twenty-nine patients (21.9%) died during the follow-up period, but only six (4.5%) due to esophageal cancer, and five (4.2%) within 6 months after resection due to other causes. The overall death-free survival rate at 5 years was 60.3%. Survival rate at 5 years was 86% for esophageal-cancer-related deaths. The overall nodal or metastatic recurrence-free survival rate at 5 years was 88.0%. Of note, among patients with no cancer recurrence nor death, median follow-up was 22 months (range 6–114). Finally, the overall recurrence rate was 16.8% (23/137), the local recurrence rate was 12.4% (17/137) (Appendix A), and the nodal or metastatic recurrence rate was 8% (11/137).

No adjuvant therapy was delivered after resection for 76 (55.5%) tumors, and 13.7% (10/76) of tumors had nodal recurrence during the follow-up. Adjuvant treatment was administered for 61 (44.5%) tumors: 9 (6.6%) had external radiotherapy, 3 (2.2%) chemotherapy, 34 (24.8%) chemo-radiotherapy, and 15 (10.9%) surgery. One patient had nodal recurrence after chemotherapy. This patient refused surgery and was not eligible for complementary radiotherapy because of a history of external radiotherapy. Among the patients who underwent complementary surgery, none had positive nodal metastasis and two had positive SCCs on esophagectomy piece (both surgery were secondary to positive margins after ER). The characteristics of adjuvant treatment by CRT or surgery are listed in Table 2. A comparison of adjuvant surgery and CRT is shown in Appendix A.

The nodal and metastatic recurrence-free survival rate at 5 years was 97.9% with and 79.1% without adjuvant therapy (*p* = 0.011, Figure 2). The nodal and metastatic recurrence-free survival rate at 5 years was 100% after adjuvant surgery and 97.1% after CRT (*p* = 0.542, Appendix A). 

### 3.3. Secondary Outcomes

#### 3.3.1. Risk Factors for Nodal or Metastatic Cancer Recurrence

In univariate analysis, age (*p* = 0.007) and the absence of adjuvant treatment by CRT or surgery (*p* = 0.011) (Table 3) were associated with nodal or metastatic cancer recurrence. In multivariate analysis, age (HR= 1.075, 95% CI: 1.007–1.149, *p* = 0.031), tumor infiltration depth > sm1 (HR = 4.129, 95% CI: 1.067–15.977, *p* = 0.040), and the absence of adjuvant CRT or surgery (HR = 11.322, 95% CI: 1.281–100.033, *p* = 0.029) were risk factors for nodal or metastatic recurrence. The use of adjuvant therapy was influenced by age: elderly patients (> 72 years) received adjuvant treatment less frequently than younger patients (*p* = 0.044) (Appendix A). A secondary analysis, excluding patients with adjuvant surgery, was performed (Appendix A). Risk factors for nodal or metastatic recurrence in that population were similar.

#### 3.3.2. Adjuvant Treatment and Risk of Nodal or Metastatic Recurrence

Among all tumors, 83 (60.6%) presented with high-risk features of lymph node invasion and 54 (39.4%) with low-risk features (Figure 3). In the high-risk group, 44 (53.0%) received adjuvant therapy, as compared to 17 (31.5%) in the low-risk group. In the high risk group, nodal recurrence-free survival rate at 5 years was 97.1% for patients who received adjuvant treatment, vs. 69.1% for those who did not (*p* = 0.008). In the low-risk group, this difference was not significant (92.0% vs. 100.0%, *p* = 0.314) (Figure 4). 

#### 3.3.3. Complications after Adjuvant Treatment

Nine severe adverse events (6.5%) were reported after adjuvant therapy. One severe infection, two esophageal strictures, and four severe esophagitis occurred after adjuvant treatment by external radiotherapy. After surgery, one fistula and one esophageal stricture were reported during follow-up. The overall survival rate at 5 years was 70% after adjuvant surgery and 61.3% after CRT (*p* = 0.872).

## 4. Discussion

This multicenter Western study is one of the largest published to date, assessing long-term outcomes of adjuvant treatment after non-curative ER for superficial SCC. Herein we report that the nodal or metastatic recurrence rate was 8% and the recurrence-free survival rate was 88% at 5 years. A tumor infiltration depth of > sm1 and the absence of adjuvant CRT or surgery were independently associated with nodal or metastatic recurrence.

Surgery is the recommended adjuvant therapy for tumors with risk of nodal invasion (≥pT1b). This is the first study to validate the ESGE expanded definition of ER infiltration of ≤ 200 µm [4]. Adjuvant treatment was not necessary in the low-risk group of lymph node invasion (infiltration depth ≤ sm1, without lymphovascular invasion, and well differentiated) because it did not significantly reduce the risk of nodal or metastatic recurrence (*p* = 0.314). The nodal or metastatic recurrence-free survival rate at 5 years was 92.0% in the low-risk group without adjuvant treatment. However, adjuvant treatment significantly reduced the risk of nodal or metastatic recurrence in the high-risk group of lymph node invasion. The nodal or metastatic recurrence-free survival rate at 5 years was 97.1% in the high-risk group with adjuvant therapy. Furthermore, the risks of nodal or metastatic recurrence were similar after adjuvant CRT or surgery (*p* = 0.542).

This study confirms the findings of previous studies [13,14,15,16,19]. ER combined with CRT is an organ preservation strategy for tumors with pejorative histoprognostic criteria and high risk of lymph node involvement [22]. The organ preservation strategy of combined ER and adjuvant treatment can be an alternative to esophagectomy with a similar disease-free survival rate [17]. Exclusive CRT is one of the less-invasive alternatives [12]. However, radiotherapy is associated with a high risk of locoregional recurrence [23]. Combination therapy with ER followed by adjuvant CRT reduces the risk of nodal recurrence [24]. This organ preservation strategy may be considered as an alternative to surgery for patients with severe comorbidities treated endoscopically for superficial SCC with a high risk of nodal invasion.

In our study, patients aged >72 years received adjuvant treatment less frequently than younger patients (Appendix A). The presence of co-morbidities did not significantly influence the choice of adjuvant treatment (Table 2 and Appendix A). CRT is one of the less-invasive alternative treatments, as opposed to esophagectomy. Mortality after esophagectomy is around 4.0% in expert centers, but this rate is higher than 10.0% in non-expert centers [11,12]. In this study, the rates of complications after adjuvant treatment were similar between surgery and CRT. The overall death-free survival rates were also similar after adjuvant surgery or CRT.

This study has several limitations. First, because of its retrospective design. Some patients were lost during follow-up. In order to limit this bias, we only included patients with a minimum of 6 months follow-up, thus excluding cases of early nodal or metastatic recurrence or death. Second, approximately half of the ER were performed by EMR. The ESGE guidelines recommended performing ESD as the first line therapy for SCC because the disease recurrence rate after EMR resection ranges from 2.4% to 26% [25,26,27,28,29]. The rates of recurrence after ESD resection are reportedly 0.0% to 7.1% [30,31,32,33]. The total number of nodal or metastatic recurrence was only 8% (11/137). Thus, the statistical power was weak for solid results. Furthermore, there was no standard strategy for the follow-up or oncological treatment: oncological strategy was discussed locally in a multidisciplinary meeting.

## 5. Conclusions

This work shows that adjuvant treatment was not necessary in patients at low risk of lymph node invasion (infiltration depth ≤ sm1, without lymphovascular invasion, and well differentiated). For patients at high risk of lymph node invasion, adjuvant treatment reduced the risk of nodal or metastatic recurrence. Organ preservation strategy associating ER and adjuvant treatment can be an alternative to esophagectomy with a similar disease-free survival rate. Further prospective randomized studies are needed to confirm these results and to better define the organ preservation strategy after ER for tumors at high risk of lymph node involvement.

## Figures and Tables

**Figure 1 cancers-15-00590-f001:**
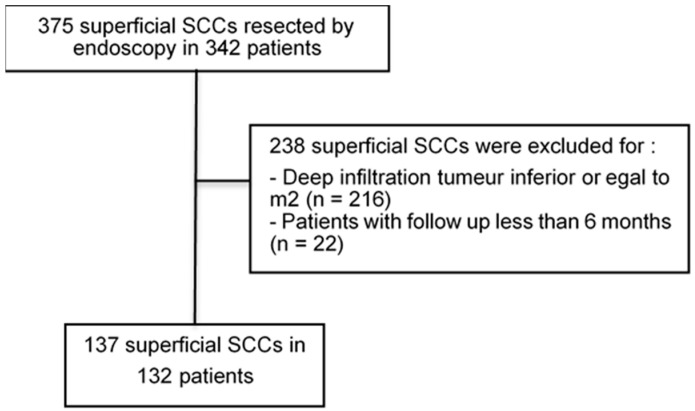
Flow chart of tumors that underwent esophageal endoscopic resection. SCC, squamous cell carcinoma; m2, intramucosal invasive carcinoma limited to the lamina propria mucosae.

**Figure 2 cancers-15-00590-f002:**
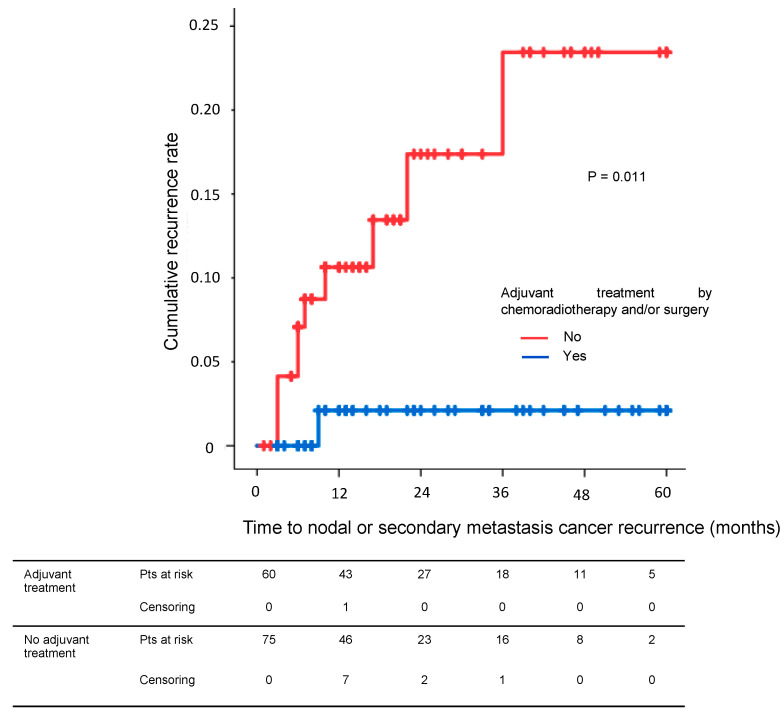
Nodal or metastatic recurrence-free survival rates in patients with tumors invading the muscularis mucosae (pT1a-m3) or the submucosa (pT1b-Sm) with and without adjuvant chemoradiotherapy and/or surgery.

**Figure 3 cancers-15-00590-f003:**
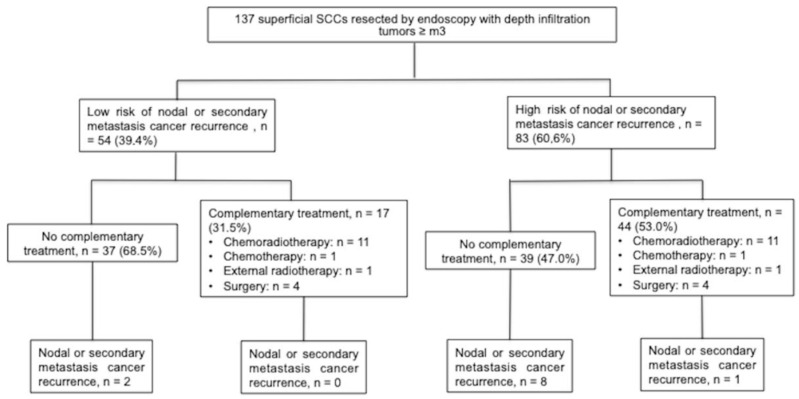
Flow chart of tumors after endoscopic resection according to the risk of nodal or metastatic recurrence with and without adjuvant chemoradiotherapy and/or surgery. SCC, squamous cell carcinoma; m3, invasive carcinoma limited to the muscularis mucosae; sm, invasive carcinoma limited to the submucosa. Low risk of nodal and/or distal metastasis was defined as an infiltration depth of ≤Sm1, with no lymphovascular invasion and well differentiated. A high risk of nodal and/or distal metastasis was defined as an infiltration depth of ≥Sm2, positive lymphovascular invasion, or mildly and poorly differentiated.

**Figure 4 cancers-15-00590-f004:**
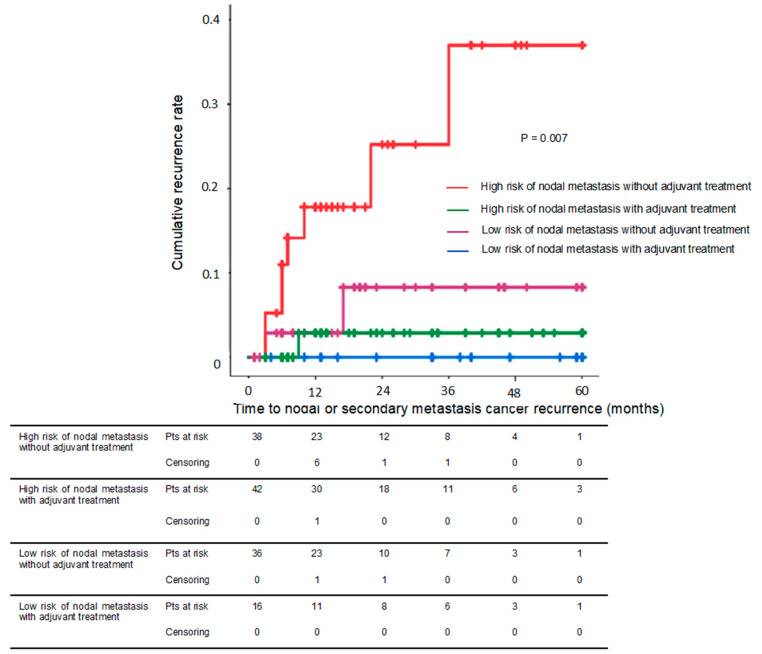
Metastatic recurrence-free survival rates in patients at low or high risk of nodal or distal metastasis with and without adjuvant chemoradiotherapy and/or surgery. SCC, squamous cell carcinoma. Low risk of nodal and distal metastasis was defined as an infiltration depth of ≤ Sm1, with no lymphovascular invasion and well differentiated. A high risk of nodal and/or distal metastasis was defined as an infiltration depth of ≥Sm2, positive lymphovascular invasion, or mildly and poorly differentiated.

**Table 1 cancers-15-00590-t001:** Tumor characteristics (*n* = 137).

Patients	Overall (*n* = 137)
Male *n* (%)	103 (75.2)
Median age (years, (range))	63 (35–90)
ASA score ≥ 3 *n* (%)	71 (51.8)
Tumor	
Median size (mm, (range))	25 (3.0–90.0)
Circumference of the esophageal lumen (%) (<1/3; 1/3–3/4; >3/4)	38.7; 39.4; 21.9
Resections *n* (%)	
Method of endoscopic resection	
- EMR	66 (48.2)
- ESD	71 (51.8)
Histology *n* (%)	
SCC	137 (100)
Lateral margin-free cancer (%) (R0; R1; Rx) ^a^	51.1; 12.4; 36.5
Cancer infiltration depth	
- m3	56 (40.9)
- sm1	35 (25.5)
- sm2–3	46 (33.6)
Differentiation *n* (%)	
- G1 ^b^	95 (69.3)
- G2 ^b^ and G3 ^b^	42 (30.7)
Lymphovascular invasion *n* (%)	19 (13.9)

ASA, American Society of Anesthesiologists; ESD, endoscopic submucosal dissection; EMR, endoscopic mucosal resection; m3, invasive carcinoma limited to the muscularis mucosae; sm1, invasive carcinoma limited to the submucosa at <200 μm, sm2–3, invasive carcinoma limited to the submucosa at >200 μm. ^a^ Lateral margins were considered R0 if negative for tumor, R1 if positive for tumor, and Rx when the pathology was not evaluable, for example, due to piecemeal resection. ^b^ G1 was well differentiated, G2 was moderately differentiated, and G3 was poorly differentiated.

**Table 2 cancers-15-00590-t002:** Characteristics of adjuvant chemoradiotherapy or/and surgery.

Patient Characteristics	Complementary Treatment	
	Yes (*n* = 61)	No (*n* = 76)
Male, *n* (%)	47 (77.0)	56 (73.7)
Age, mean (range), years	64.7 (44–90)	62.8 (35–83)
ASA score ≥3, *n* (%)	32 (52.5)	39 (51.3)
Tumor size, median (range), mm	25 (5–81)	20 (3–90)
EMR resection, *n* (%)	30 (49.2)	41(53.9)
Tumor infiltration depth m3–sm, *n* (%)	34 (55.7)	57 (75.0)
Differentiation G2 and G3 ^†^, *n* (%)	22 (36.1)	20 (26.3)
Lymphovascular invasion, *n* (%)	11 (18.0)	8 (10.5)

ASA, American Society of Anesthesiologists; EMR, endoscopic mucosal resection; m3, invasive carcinoma limited to the muscularis mucosae; sm, invasive carcinoma limited to the submucosa. ^†^ G2 was moderately differentiated and G3 was poorly differentiated.

**Table 3 cancers-15-00590-t003:** Risk factors associated with nodal or distal metastasis recurrence during follow-up.

Risk Factors to Nodal or Distal Metastasis	Nodal or Secondary Metastasis Cancer Recurrence		Univariate Analysis	Multivariate Analysis
	Yes (*n* = 11)	No (*n* = 126)	*p*-Value ^1^	Hazard Ratio (95% CI)
Male *n* (%)	8 (72.7)	95 (75.5)	0.691	
Mean age, (years, (range))	70.2 (56–83)	63.3 (35–90)	0.007	1.075 (1.007–1.149)
History of head-and-neck cancer *n* (%)	4 (36.4)	51 (40.5)	1.000	
Median tumor size (mm, (range))	20.0 (7.0–71.5)	25.0 (3.0–90.0)	0.827	
Tumor circumference of the esophageal lumen > 1/3 *n* (%)	4 (36.4)	80 (63.5)	0.096	
EMR resection *n* (%)	7 (63.6)	64 (50.8)	0.667	
Piecemeal resection *n* (%)	6 (54.5)	44 (34.9)	0.295	
R1 or Rx ^a^ lateral margins, *n* (%)	6 (54.5)	61 (48.4)	0.901	
Rx ^a^ depth margins, *n* (%)	3 (27.3)	24 (19.0)	0.570	
Tumor infiltration depth > Sm1 *n* (%)	6 (54.5)	40 (31.7)	0.145	4.129 (1.067–15.977)
Differentiation G2 and G3 ^†^, *n* (%)	6 (54.5)	36 (28.6)	0.067	
Lymphovascular invasion, *n* (%)	1 (9.1)	18 (15.7)	0.677	
No adjuvant treatment by chemoradiotherapy and/or surgery *n* (%)	1 (9.1)	60 (47.6)	0.011	11.322 (1.281–100.033)
Median follow-up (months, (range))	34.0 (9–112)	21.5 (1–135)	0.547	

CI, confidence interval; EMR, endoscopic mucosal resection; sm1, cancer invading the submucosa; R1, margin not tumor free. ^a^ R1 for microscopic margins positive for tumor and Rx when the pathology was not evaluable, for example, due to piecemeal resection and/or coagulation artifacts. ^†^ G2 was moderately differentiated and G3 was poorly differentiated. ^1^ Univariate analyses were carried out by log-rank test for qualitative variables. For age and tumor size, a univariate Cox model was used. All significant factors, together with those of borderline significance (*p* < 0.2), were included in the multivariate analyses, which used the Cox proportional hazards model. In the multivariate models, only variables with a *p* < 0.05 were included.

## Data Availability

The data presented in this study are available on request from the corresponding author.

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
