# Peer review of "Efficacy of Organ Preservation Strategy by Adjuvant Chemoradiotherapy after Non-Curative Endoscopic Resection for Superficial SCC: A Multicenter Western Study"

_cancers, 2023, doi:10.3390/cancers15030590_

Round 1

Reviewer 1 Report

This work is an interesting and comprehensive summary for the understanding of the treated topic. The paper is written well. The abstract provides adequate information on the discussion of the following paragraphs. The introduction is comprehensive and offers an adequate background of knowledge on the treated topic. The objectives and outcomes of the work are clearly described. However, I'd like to point out several weak points in this manuscript:

  • The Materials and Methods should be described with sufficient details to allow others to replicate and build on the published results. New methods and protocols should be described in detail while well-established methods can be briefly described and appropriately cited.
  • It is necessary to increase the number of patients involved in the study for greater significance of the results.
  • This study is not new. In fact, a recently published paper describes a similar study. Therefore, it is suggested to add it in the references section. The reference is: Sakiko Naito et al.  Long-term outcomes of esophageal squamous cell carcinoma with invasion depth of pathological T1a-muscularis mucosae and T1b-submucosa by endoscopic resection followed by appropriate additional treatment. Digestive endoscopy 2022 May;34(4):793-804. doi: 10.1111/den.14154. Epub 2021 Oct 20.
  • Re-reading of the manuscript is suggested to avoid errors. Review the figures numbering to avoid repeating the same number.

Author Response

Response to reviewer 1 :

  1. Materials and Methods should be described with sufficient details to allow others to replicate and build on the published results. New methods and protocols should be described in detail while well-established methods can be briefly described and appropriately cited.

We kindly thank the Reviewer for this comment. CRT protocol was detailed in the supplementary appendix before reviewing. We have therefore added the CRT protocol to the main manuscript in the “Method” section following your advice. It reads as follows:

“CRT were carried out depending on each patient’s comorbidity background. The CRT protocol was in accordance with French guidelines (23). Radiation treatment was delivered at 1.8–2.0 Gy per fraction, with a total dose of 40–50 Gy over 25–30 fractions. Chemotherapy was chosen between the two following regimens and delivered during radiotherapy at 3-weeks intervals:

  • 5-fluorouracil (800 mg/m2) administered on days 1 to 5, in association with Cisplatin (100 mg/m2) administered with hydration on day 1.
  • Or 5-fluorouracil (400 mg/m²) in bolus on day 1 followed by 5-fluorouracil (1600 mg/m2) administered over 48H, in association with Oxaliplatine (85 mg/m2) with hydration and folic acid on day 1. “
  1. It is necessary to increase the number of patients involved in the study for greater significance of the results.

We agree with the Reviewer. Unfortunately, the retrospective design of this study did not allow for more patients. However, we think that it warrants further prospective randomized studies to confirm these results on the one hand, and on the other hand to better define the organ preservation strategy after ER for tumors at high risk of lymph node involvement.

  1. This study is not new. In fact, a recently published paper describes a similar study. Therefore, it is suggested to add it in the references section. The reference is: Sakiko Naito et al. Long-term outcomes of esophageal squamous cell carcinoma with invasion depth of pathological T1a-muscularis mucosae and T1b-submucosa by endoscopic resection followed by appropriate additional treatment. Digestive endoscopy 2022 May;34(4):793-804. doi: 10.1111/den.14154. Epub 2021 Oct 20.

We once again agree with the Reviewer. Currently, most important studies come from Eastern countries and there is limited data on this topic in Western populations. This, according to us, highlights the importance of displaying these results. We have added recent relevant papers in our Introduction, Discussion and Reference sections.

  1. Re-reading of the manuscript is suggested to avoid errors. Review the figures numbering to avoid repeating the same number.

We thank the Reviewer for this comment and have addressed these elements in the manuscript.

Reviewer 2 Report

With great interest I’ve read the article by Arthur Berger and colleagues on the efficacy of neoadjuvant chemoradiotherapy (CRT) in patients after non-curative treatment for superficial esophageal squamous cell carcinoma (ESCC).

The study is a retrospective analysis performed within 11 French tertiary care hospitals (period 04.1999 - 04.2018), most of which are centers of excellence in endoscopic therapy with a long track record of publications on the topic. The study’s primary aim was to compare the long-term outcomes (5-years OS, 5-year recurrence-free survival) between patients who received and did not receive adjuvant treatment after non-curative endoscopic resection for superficial ESCC.

The study is of great importance as it describes a new approach in the treatment of ESCC and is one of the most extensive reports on a Western population. However, I have a few major and minor comments regarding the study.

Major comments:

·   As the authors claim in the Introduction, CRT following non-curative endoscopic resection of ESCCs could be a new, less-invasive, “organ-preservation” treatment strategy. So why did the authors include patients who actually underwent surgery (n=15[10.9%])? I would advise removing patients who received adjuvant surgical treatment in this analysis.

·         Please specify the CRT adjuvant treatment regime – was it decreased doses of radiation for “prophylactic” treatment or full Rth doses as in definitive CRT? What Cth were usually given? Also, could you elaborate on the actual patterns of decision-making why did some patients received and others did not receive adjuvant CRT?

·         MATERIALS AND METHODS (Page 2; Line 82-96): the first three paragraphs seem to be copied from the “Manuscript Preparation” instructions on the publisher’s website. Please remove this.

·         DISCUSSION: (Page 270-273): Similarly to the problem above, the first paragraph is a copied text from the publishers’ website – please remove it.

·         The treatment of choice for ESCC is ESD; however, over half of your patients had a resection performed via EMR (51.8%). Could you explain why (for example, was it more often performed in the older years of the study period, e.g., before 2010) and whether this influenced the patients’ outcomes?

·         In addition to comment above: the study covers a very long time-period (1999-2018). Many things have changed in medicine during that time. You could consider adding the treatment period (e.g., before 2010 vs. after 2010) in the multivariable analysis to see whether this was significant factor.

Minor comments:

·         A very high fraction of patients from your cohort (41.6%) had a previous head-and-neck cancer(HNC) – Firstly, did you consider including this factor in the multivariable model? Secondly, was this a significant factor when deciding on the adjuvant CRT? After all, many of these patients had previous radiation, which is often a discouraging factor for another CRT treatment. In other words, how many of the patients who did not receive adjuvant CRT had previous HNC?

·         How many patients had multiple tumors in the esophagus (>1 SCC tumor) at presentation, and how did you handle this data in the analysis?

·         Page 3; Line 128: “mild and poor different-…“ – you meant “moderate”?

·         Page 3; Line 141: typo. Change “repeted” to “repeated” (or “performed”).

Round 2

Reviewer 2 Report

I thank the authors for considering my suggestions and for their hard work in refining the article. I am happy with the current version of the manuscript.